# Morpho-agronomic, biochemical and molecular analysis of genetic diversity in the Mesoamerican common bean panel

Alison Fernando Nogueira[1], Vania Moda-Cirino[2], Jessica Delfini[1], Luriam Aparecida Brandão[1], Silas Mian[1], Leonel Vinicius Constantino[1], Douglas Mariani Zeffa[2], José dos Santos Neto[2], Leandro Simões Azeredo Gonçalves[1]*

1 Agronomy Department, Universidade Estadual de Londrina (UEL), Londrina, Paraná, Brazil, 2 Instituto de Desenvolvimento Rural do Paraná – IAPAR – EMATER (IDR–IAPAR–EMATER), Londrina, Paraná, Brazil

* leandrosag@uel.br

**Data Availability Statement:** All relevant data are within the manuscript and its Supporting information files.

## Abstract

The common bean (*Phaseolus vulgaris* L.) is of great importance to the food and nutritional security of many populations, and exploitation of the crop's genetic diversity is essential for the success of breeding programs. Thus, the aim of the present study was to evaluate the genetic diversity of 215 common bean accessions, which included cultivars, obsolete cultivars, improved lines, and landraces using morpho-agronomic and biochemical traits, and amplified fragment length polymorphism markers (AFLP). Genetic parameters, box plots, Pearson's correlation analysis, and Ward's hierarchical clustering were used to analyze the data. The Jaccard similarity coefficient and neighbour-joining clustering method were used for molecular analysis. A wide variability among the accessions was observed for morpho-agronomic and biochemical traits. Selective accuracy (Ac) and broad-sense heritability ($h^2$) values were high to intermediate for all traits, except seed yield. Ward's hierarchical clustering analysis generated six groups. AFLP analysis also revealed significant differences among the accessions. There was no correlation between the differences based on genetic markers and those based on morpho-agronomic and biochemical data, which indicates that both datasets are important for elucidating the differences among accessions. The results of the present study indicate great genetic diversity among the evaluated accessions.

## Introduction

The common bean (*Phaseolus vulgaris* L., Fabaceae) is one of the world's most important legume crops and is a valuable source of dietary protein, fiber (soluble and insoluble), minerals (calcium, potassium, magnesium, iron, zinc, manganese, and copper), and vitamins (especially B-complex vitamins) worldwide [1]. The species also contains a large amount of genetic variation, and two main gene pools (Andean and Mesoamerican) are generally recognized, based on their domestication [2,3]. Mesoamerican cultivars are characterized by small (<25 g per 100) or medium (25–40 g per 100) seeds and "S" or "B" phaseolin patterns, whereas Andean cultivars are characterized by large seeds (>40 g per 100) and "T", "C", "H", or "A" phaseolin

**Funding:** The author(s) received no specific funding for this work.

**Competing interests:** The authors have declared that no competing interests exist.

patterns [4], and different regions of the world exhibit preferences for beans from one gene pool or the other. More specifically, Mesoamerican cultivars are more popular in North America, Central America, and parts of South America, whereas Andean cultivars are more popular in Africa, Europe, and other parts of South America [5]. In Brazil, which is one of the world's largest producers and consumers of beans, Mesoamerican cultivars are more popular, and carioca and black commercial types are preferred, accounting for 85 and 11% of Brazil's bean production, respectively, which is approximately three million tons of grain [6,7].

Over the last few decades, the yield of bean crops in Brazil has increased significantly [6]. This improvement has largely been attributed to genetic improvement, which has increased the yields of carioca and black beans by 0.72–6.74% and 1.10–2.42%, respectively [7,8]. However, simple sequence repeat (SSR) marker analysis by Delfini et al. [9] revealed that the main Brazilian bean cultivars possess relatively low genetic diversity. In such cases, the introduction of new accessions to crop breeding programs is considered an important strategy for improving genetic diversity, and accessions with valuable traits, such as nutritional quality and resistance or tolerance to abiotic and biotic stresses, are often preserved in gene banks [10]. However, to be used in breeding programs, the accessions must be fully evaluated and characterized, both morphologically and molecularly.

The precise exploration of genetic diversity by the breeding programs is necessary for the development of new commercial cultivars adapted to the most diverse regions of the country. Usually, higher levels of diversity in the set used for breeding is related to a greater chance of identifying traits of agronomic interest [11,12]. The common bean exhibits wide agronomic traits variation, including cycle (early and late), growth habit (determinate and indeterminate), plant habit (erect, semi-erect, and prostrated), plant morphology, and seed characteristics (shape, size, color, biochemical composition, and functional composition) [12,13].

The common bean gene bank of the Instituto de Desenvolvimento Rural do Paraná—IAPAR–EMATER (IDR–Paraná) maintains a collection of 14,164 accessions, which include landraces, lines improved by the institute itself, introductions from other research institutions and universities, and both obsolete and current cultivars from several breeding programs [14,15]. Accordingly, the aim of the present study was to analyze the genetic diversity of Mesoamerican bean accessions from the IDR–Paraná gene bank and to use morpho-agronomic, biochemical, and molecular traits to select accessions with superior traits.

## Material and methods

### Plant material

The Mesoamerican Panel of Bean Diversity (MPBD), which is a collection of 215 common bean accessions in the IDR–Paraná gene bank (S1 Table), was selected for evaluation. Among these accessions, there are landraces, improved lines, and obsolete and modern cultivars of different commercial groups (carioca, black, and colored) from different breeding programs.

### Experimental design

The experiment was conducted during the 2018–2019 rainy crop season at the IDR–Paraná Research Station in Londrina, Paraná, Brazil (Latitude: 23˚17' S, Longitude: 51˚10' W, and altitude 550 m). The experiment was conducted using a Federer's augmented block design [16] and four commercial cultivars as checks, two carioca (C1: IPR Quero-quero and C2: IPR Campos Gerais) and two black cultivars (C3: IPR Urutau and C4: IPR Tuiuiú). Each plot included four 2-m-long rows with 0.5 m spacing between rows and 12 plants per meter, the two central rows making up the useful area. Base fertilization was carried out according to the results of soil chemical analysis, and nitrogen top-dressing fertilization (200 kg ha$^{-1}$ of ammonium

sulfate) was applied during the $V_4$ development stage [17]. Chemical control of pests, diseases, and invasive plants was performed when necessary using products registered for the crop.

## Morpho-agronomic characterization

Seven uniform and representative plants were collected at physiological maturation ($R^9$ development stage) from each experimental plot. The following morpho-agronomic traits were evaluated: stem length (STL, in cm), insertion height of the first pod (IFP, in cm), number of nodes on main stem (NN), pod length (PL, in cm), number of seeds per pod (SP), and weight of 100 seeds (W100). Seed yield (YLD, kg ha$^{-1}$ with moisture of 13%) was obtained after manual removal and mechanical threshing of plants from the two central rows of each plot.

## Biochemical characterization

The following biochemical traits were evaluated: total phenolic content (TPC), total flavonoid content (TFC) and antioxidant (DPPH-scavenging) activity (DPPH). For these analyzes, fifty-seed samples were taken from the beans harvested from each experimental plot, ground using a Willey MA340 type knife mill (Marconi Laboratory Equipment, Piracicaba, Brazil), passed through a 60-mesh sieve, packed, and stored at –18˚C. Moisture content (%) was measured in triplicate by drying bean flour samples (2 g) at 105 ± 3˚C until achieving constant weight [18].

To measure FT, FLA, and DPPH, extracts were prepared from each fresh sample (1.0 g) using 10 mL ethanol (70%, v/v). Briefly, each suspension was shaken for 2 h at room temperature (~28˚C), centrifuged at 1013 × g for 5 min, and then filtered through cotton fabric [19].

To measure FT, 1.0 mL methanolic extract was mixed with 1.0 mL methanol, 1.0 mL Folin-Ciocalteu reagent (0.2 N), and 1.0 mL sodium carbonate (10%, w/v) and incubated at 25˚C for 30 min in the dark. Subsequently, absorbance at 765 nm was measured using a AJX-1600 spectrophotometer (Micronal,). Gallic acid (10.0–100.0 mg L$^{-1}$) (r = 0,9960), was used as the standard, and the results were expressed as mg of gallic acid equivalent per 100 g dry sample (mg GAE/100 g; [20].

To measure FLA, 1.0 mL methanolic extract was mixed with 1.0 mL aluminum chloride (5.0%, w/v) and 2.0 mL methanol and incubated for 30 min in the dark. Subsequently, absorbance at 425 nm was measured using a spectrophotometer (Micronal, AJX-1600). Quercetin (50.0–500.0 mg L$^{-1}$) (r = 0,9942), was used as the standard, and the results were expressed as mg quercetin equivalent per 100 g dry sample (mg QE 100 g$^{-1}$; [21]). To measure DPPH, 50.0 μL metabolic extract was mixed with 1.0 mL acetate buffer (100 mM, pH 5.5), 1.0 mL methanol, and 0.5 mL DPPH-ethanol solution (250.0 μM) and then incubated at room temperature (~28˚C) for 15 min in the dark. Subsequently, absorbance at 517 nm was measured three times using a spectrophotometer (Thermo Fisher Scientific, Massachusetts, USA). Trolox (6-hydroxy-2,5,7,8-tetramethylchroman-2-carboxylic acid) (r = 0,9992)was used as the standard, and the results were expressed as μmol Trolox equivalent antioxidant capacity per 100 g dry sample (μmol TEAC 100 g$^{-1}$; [22].

## Molecular characterization

Leaves were sampled from young plants and DNA was extracted using a modified version of the protocol of Ferreira and Grattapaglia [23], with the use of CTAB buffer followed by isopropanol precipitation. After extraction, all samples were treated with RNAse (110 ng ml$^{-1}$), and DNA integrity was confirmed using 1% agarose gel electrophoresis, whereas concentration and purity were determined by spectrophotometry using a NanoDrop 2000/2000c (Thermo Fisher Scientific, Massachusetts, USA). Only the DNA samples with A260/280 nm ratios between 1.8 and 2.2 were used for further analysis.

Amplified fragment length polymorphism (AFLP) markers were amplified as described by Vos et al. [24], with modifications. Briefly, DNA from each accession was double digested by incubating the DNA (~700 ng) in 20 μL reactions with EcoRI and MseI (5 U each) and 2 μL MseI 10X assay buffer at 37˚C for 18 h. The resulting fragments were ligated to EcoRI (0.5 μM) and MseI (5 μM) adaptors by incubating the fragments in 10 μL reactions with T4 DNA ligase (1 U), T4 DNA ligase 1X buffer, NaCl (0.05 M), BSA (50 μg μL$^{-1}$), and DTT (0.25 mM) at 37˚C for 3 h, 17˚C for 30 min, and 70˚ C for 10 min. Digestion and ligation were confirmed using 1% agarose gel electrophoresis, and successfully ligated sample fragments were diluted 1:4 in ultrapure water.

Subsequently, the fragments were amplified with a pair of pre-selective primers containing a selective nucleotide. Pre-selective amplification was performed in 10-μL reactions that contained 3.5 μL GoTaq Green Master Mix$^{®}$ kit, 0.58 uL pre-selective primer (4.75 μM), and 3.0 μL diluted restriction/ligation product. The reactions were subject to the following thermal cycler program: 2 min at 72˚C; followed by 20 cycles of 1 s at 94˚C, 30 s at 56˚C, 2 min at 72˚C; and 30 min at 60˚C. The pre-selective PCR was confirmed using 2% agarose gel electrophoresis, and the amplified product was diluted 1:16 in ultrapure water.

Selective amplification was performed in 10 μL reactions that contained 2.5 μL diluted pre-selective product, 0.54 μL MseI (5 μM) and EcoRI (1 μM) selective primers, and 3.5 μL GoTaq Green Master Mix (Promega). The reactions were subject to the following thermal cycler program: 2 min at 94˚C, 30 s at 65˚C, and 2 min at 72˚C; followed by 8 cycles of 1 s at 94˚C, 30 s at 64–57˚C (decreasing 1˚C per cycle), and 2 min at 72˚C; then followed by 23 cycles of 1 s at 94˚C, 30 s at 56˚C, and 2 min at 72˚C; ending finally 30 min at 60˚C. Four combinations of the EcoRI and MseI primers (E-ACA/M-CAC, E-ACG/M-CAA, E-ACT/M-CAA, E-ACG/ M-GAC) were tested, containing three selective nucleotides, visualized in 7% polyacrylamide gel. The choice of primers was based on previous laboratory work (E-AAG/M-CTC, E-ACT/ M-CTT, E-ACA/M-CAC, and E-AGC/M-CTAG). Subsequently, the reaction was denatured at 95˚C for 3 min and then submitted to capillary electrophoresis (Applied Biosystems, Califórnia, USA).

The products of the four selective amplifications were submitted to capillary electrophoresis using corresponding combinations of primers that were each labeled with a fluorophore (FAM, NED, VIC, or PET).

The amplified samples with the labeled primers were combined in the proportion of: 1μL FAM: 2μL NED: 2μL VIC: 2 μL PET, with 3.0 μL of ultrapure water. For the sequencing run, 10 μL reactions included 1.0 μL primer mixture, 0.2 μL GeneScan *600 LIZ* Size Standard v2.0, and 8.8 μL Hi-Di formamide. The electrophoresis results were combined in a binary matrix using GeneMapper v.4.1. All amplifications were performed using a GeneAmp PCR System 9700 (Applied Biosystems, Califórnia, USA).

## Data analysis

Data analysis was carried out by the best linear unbiased predictor (BLUP) and restricted maximum likelihood (REML) methods using the software Selegen–REML/BLUP [16]. The analysis of deviance (ANADEV) was performed considering the following statistical model:

$$y = Xf + Zg + Wb + \varepsilon,$$

where *y*, *f*, *g*, *b*, and *ε* represent the data vectors of fixed effect (overall mean), genetic effects of the accessions (random), block effect (random), and random errors, respectively. *X*, *Z*, and *W* represent the incidence matrices for *f*, *g*, and *b*, respectively. The significance of all random effects from the ANADEV were verified by the likelihood ratio test (LRT) at 5% of probability.

Broad-sense heritability ($h^2$) was estimated using the following formula: $\frac{\hat{\sigma}_g^2}{\hat{\sigma}_g^2 + \hat{\sigma}_e^2}$, where where $\hat{\sigma}_g^2$ is the genotypic variance and $\hat{\sigma}_e^2$ is the residual variance. The selective accuracy (Ac) was obtained as follows: $Ac = \sqrt{\frac{1 - PEV}{\hat{\sigma}_g^2}}$, where $PEV$ is the variance of the prediction error of the genotypic values and $\hat{\sigma}_g^2$ is the genotypic variance.

The accessions were separated according to two criteria, namely commercial group (black, carioca, and colored) and genetic material (landraces, improved lines, and cultivars), and box plots were used to compare the distributions of the groups. Pearson's correlation analysis and Ward's hierarchical clustering using Euclidean distance were also performed through BLUP values. These analyses were performed using software R [25] through the 'ggplot2' [26], 'FactoMineR' [27], 'cluster' [28], 'tidyverse' [29], 'RColorBrewer' [30], and 'corrplot' [31] packages.

For the AFLP marker data, a Jaccard distance matrix was calculated, and neighbor-joining clustering analysis was performed. The analyses were performed using Past 3.24 [32]. The groups formation in dendrograms were established using the criterion proposed by Charrad et al. [33] and the correlation of both distance matrix (phenotipic and molecular) was performed using the Mantel [34], test, with 1,000 permutations. These analyses were performed using software R through the 'NbClust' [35] and 'ape4' [36] packages.

## Results

### Morpho-agronomic and biochemical traits

**Analysis of deviance, heritability, and correlation.** The ANADEV's revealed significant differences among the morpho-agronomic and biochemical traits of the 215 accessions ($P \leq 0.05$). The Ac values were high ($\geq 0.70$) for all traits, except for YLD (0.15) (Fig 1).

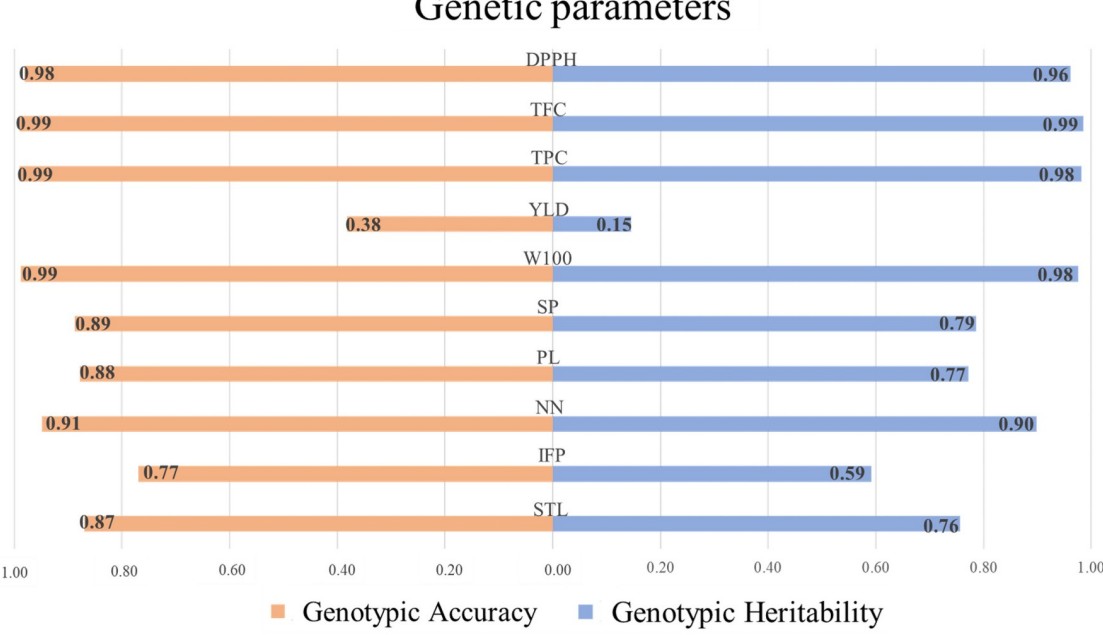

**Fig 1. Broad-sense heritability and selective accuracy for morpho-agronomic and biochemical traits in 215 Mesoamerican common bean accessions.** STL: Main stem length, IFP: Insertion height of the first pod, NN: Number of nodes on main stem, PL: Pod length, SP: Number of seeds per pod, W100: Weight of 100 seeds, YLD: Seed yield, TPC: Total phenolic contents, TFC: Total flavonoid contents, and DPPH: antioxidant capacity.

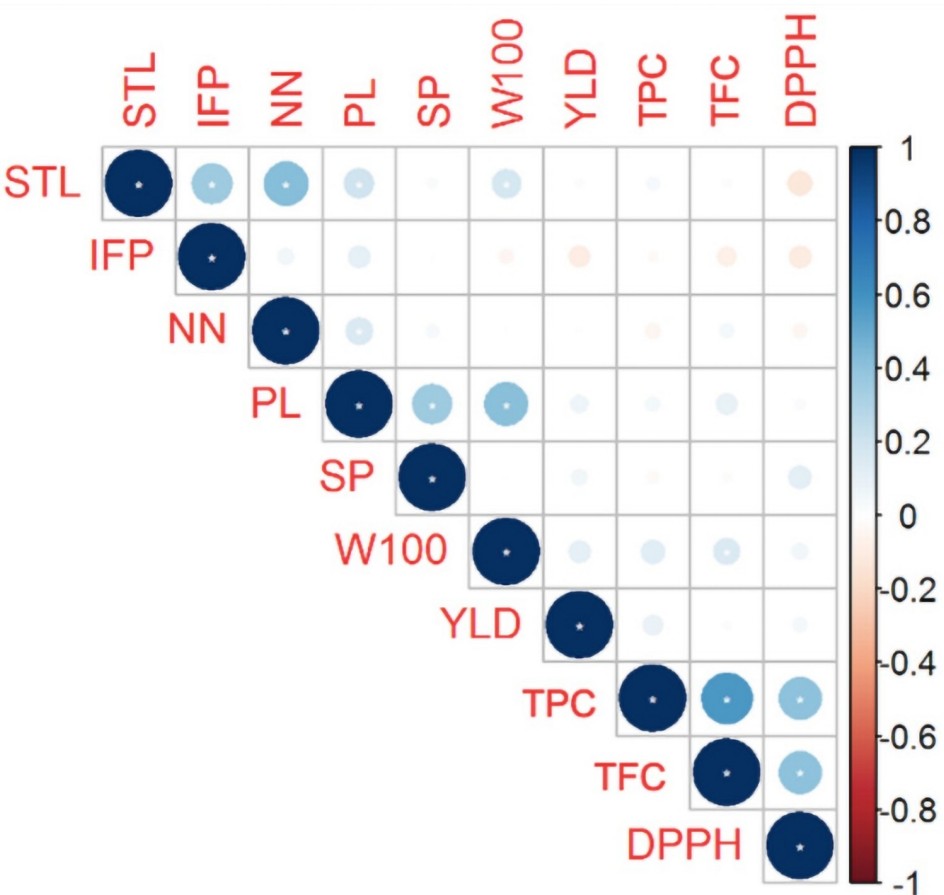

**Fig 2. Correlation between the morpho-agronomic and biochemical traits of 215 Mesoamerican common bean accessions.** STL: Main stem length, IFP: Insertion height of the first pod, NN: Number of nodes on main stem, PL: Pod length, SP: Number of seeds per pod, W100: Weight of 100 seeds, YLD: Seed yield, TPC: Total phenolic contents, TFC: Total flavonoid contents, and DPPH: Antioxidant capacity.

Meanwhile, $h^2$ was high (0.90–0.98) for DPPH, TPC, TFC, W100, and NN, intermediate (0.59–0.79) for all other traits, except for YLD (0.38).

Pearson's correlation analysis revealed significant ($P \leq 0.05$) correlations between several biochemical traits (DPPH × TPC, DPPH × TFC, and TPC × TFC; Fig 2), but correlations between these biochemical traits and morpho-agronomic traits were weak or lacking, except for the correlation between TFC and W100. Furthermore, no correlations were observed between any of the morpho-agronomic traits and YLD, and the greatest correlations among the morpho-agronomic traits were between STL × IFP, STL × NN, STL × PL, STL × W100, NN × PL, PL × SP, and PL × W100.

**Box plot.** Wide variability was observed among the MPBD accessions for all morpho-agronomic and biochemical traits (Fig 3). For the STL, IFP, NN and SP traits there was no difference in the average among the of seed color groups, while for PL, W100, TFC and DPPH the carioca group obtained the highest average value. For YLD e TPC traits, the carioca and color groups had the highest average values.

Furthermore, the landraces yielded the greatest mean IFP, NN, PL, and SP values, whereas cultivars yielded the greatest W100 and YLD values (Fig 3). However, accession type had no significant effect on STL. Regarding biochemical traits, landraces yielded the greatest mean

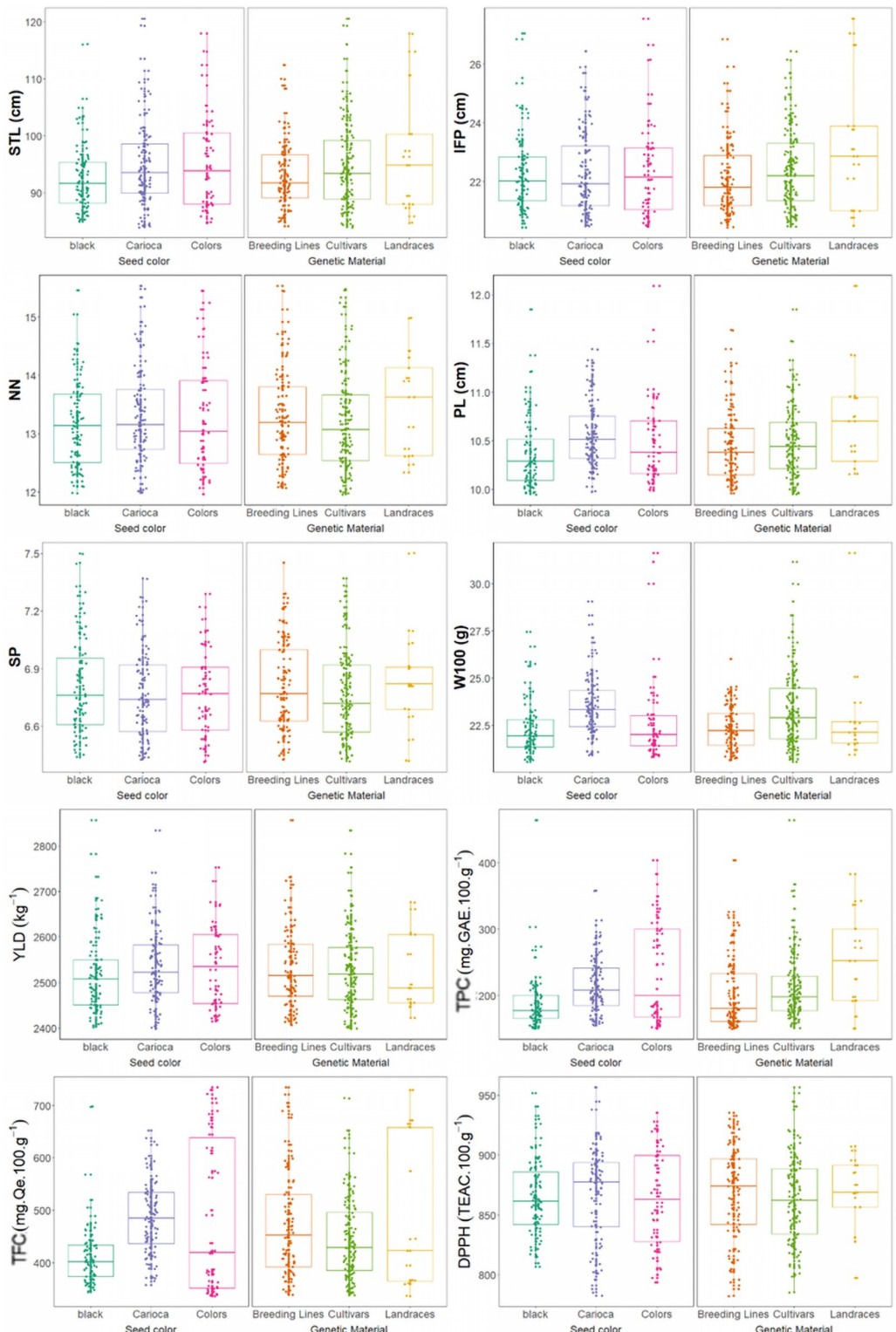

**Fig 3. Morpho-agronomic and biochemical traits of 215 Mesoamerican common bean accessions.** Box plots are based on BLUP values. STL: Main stem length, IFP: Insertion height of the first pod, NN: Number of nodes on main stem, PL: Pod length, SP: Number of seeds per pod, W100: Weight of 100 seeds, YLD: Seed yield, TPC: Total phenolic contents, TFC: Total flavonoid contents, and DPPH: Antioxidant capacity.

TPC, whereas the IDR–Paraná improvement program lines yielded the greatest TFC and DPPH.

The STL and IFP values ranged from 83.94 to 120.62 cm and 20.44 to 27.56 cm, respectively. The lowest STL values were obtained from the TAA Gol, Carioca 1070, IPR Colibri, IPR Curió, and IAPAR 57 accessions (83.00, 94.00, 84.10, 84.23, 84.36, and 84.48 cm, respectively), and the lowest IFP values were obtained from the LP33, G5285, IPR Eldorado, G2358, and LP37 accessions (20.44, 20.46, 20.48, 20.50, and 20.51 cm, respectively). In contrast, the greatest STL values were obtained from the FT 65, BRS Requinte, Rosinha G1, Ouro-Negro, and Roxo de Mato Grosso accessions (120.62, 119.37, 117.97, 116.08, and 114.82 cm, respectively), and the greatest IFP values were obtained from the Roxo de Minas, Roxinho Ivaí, NAB 87, G1261, and IAC Milênio accessions (27.56, 27.05, 26.83, 26.66, and 26.44 cm, respectively).

The greatest W100 values were obtained from the Rosinha G1, G5285, IPA 9, TAA Gol, and Pearl accessions (31.65, 31.17, 30.0, 29.07, and 28.34 g, respectively). The YLD values ranged from 2398.27 to 2857.16 kg ha$^{-1}$, and the greatest YLD values were obtained from the NAB 87, IPR Maracanã, Macanudo, IPA 6, and IPR 139 Juriti Claro accessions (2857.16, 2834.40, 2782.91, 2753.31, and 2741.54 kg ha$^{-1}$, respectively). The NAB 87, Macanudo, DOR 500, LP35, and LP43 accessions yielded the greatest YLD values among the black accessions, whereas the IPR Maracanã, IPR 139 Juriti Claro, EMP 250, LP03, and IAPAR 14 accessions yielded the greatest YLD values among the carioca accessions.

Among the biochemical traits, TPC, TFC and DPPH range from 149.62 to 464.71 mg GAE 100 g$^{-1}$, from 336 to 734.62 mg quercetin 100 g$^{-1}$, and from 782.43 to 956.69 mg TEAC 100 g$^{-1}$, respectively. The greatest TPC values were obtained from the Awauna, BAT 1192, Roxo de Mato Grosso, Sapira and Flor Diniz accessions (464.71, 403.85, 383.32, 367.80, and 357.98 mg GAE 100 g$^{-1}$, respectively), whereas the greatest TFC values were obtained from the DOR 483, Roxo de Minas, DOR 364, Rio Vermelho and MUS 49 accessions (734.62, 729.47, 721.35, 713.68, and 704.95 mg quercetin 100 g$^{-1}$, respectively), and the greatest DPPH values were obtained from the BRS Ametista, Diamante Negro, IAC Carioca Tybatã, BRS Campeiro and Aporé accessions (956.69, 951.77, 944.67, 940.70, and 937.92 mg TEAC 100 g$^{-1}$, respectively).

**Multivariate analysis.** Ward's hierarchical clustering analysis resulted in the formation of six groups by Charrad et al. [35] criterium (Fig 4). Group I contained 31 accessions, including 18 black (58%), 10 carioca (32%), and three coloreds (10%). The group yielded low mean STL and NN values, but also yielded the greatest SP. Group II contained 39 accessions, including 24 coloreds (62%), eight carioca (20%), and seven black (18%). This group also had low mean STL and NN values, as well as low W100 and TFC. Groups I and II had lower mean YLD values (2487.59 and 2476.37 kg ha$^{-1}$, respectively) than the other groups. Group III contained the highest number of accessions (n = 48), including 26 carioca (54%), 17 black (36%), and 5 coloreds (10%). The group yielded the lowest mean IFP values and high DPPH values. Group IV contained 27 accessions, including 13 coloreds (48%), 12 carioca (44%), and two black (7%). The group yielded the greatest TPC, TFC, and DPPH values. Groups V contained 43 accessions, most of which were carioca-type ($n$ = 22). Most of the Group VI and V accessions were carioca-type (44% and 51%, respectively), and Groups VI and V yielded the greatest mean STL and NN values. Group VI also yielded the greatest mean W100. Groups III, IV, V, and VI yielded mean YLD values of 2546.69, 2542.49, 2506.40, and 2592.77 kg ha$^{-1}$, respectively.

## Molecular diversity

The four AFLP primer pairs yielded a total of 973 bands, of which 921 (94.6%) were polymorphic. The E-AAG/M-CTC, E-ACT/ M-CTT, E-ACG/M-CTAG, and E-ACA/M-CAC combinations yielded 122, 141, 200, and 458 bands, respectively. The coefficient of similarity used to

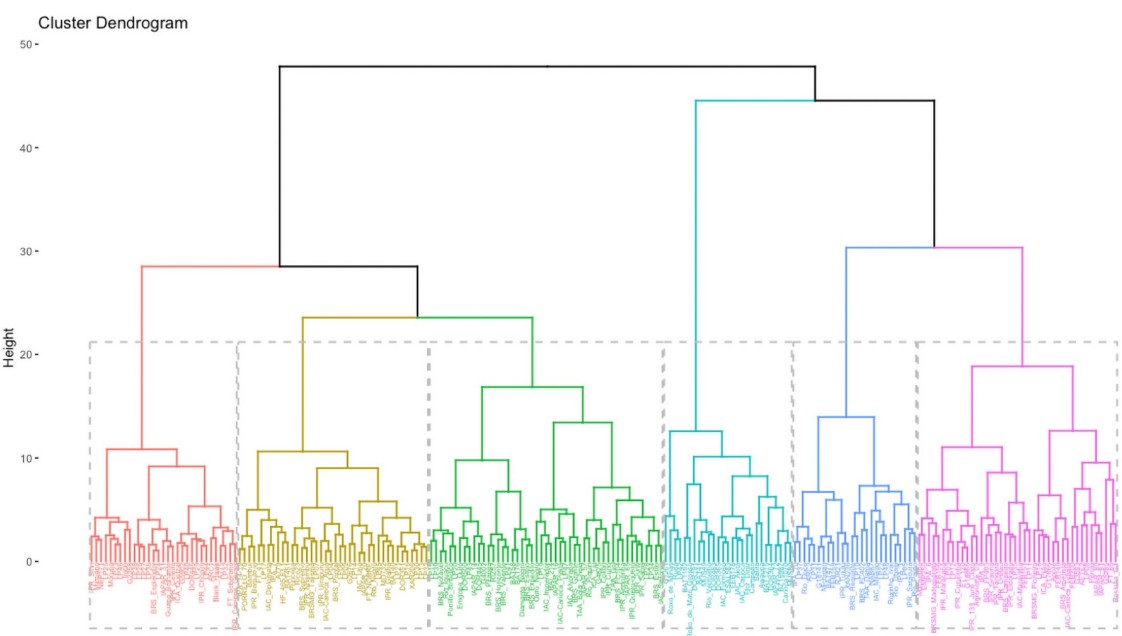

**Fig 4. Ward's hierarchical clustering analysis of 215 Mesoamerican common bean accessions.** Analysis was performed using standardized Euclidean distance.

calculate the genetic distance between the 215 accessions ranged from 0.26 (DOR364 and LP37) to 0.79 (IAC Formoso and FT 65).

Neighbour-joining hierarchical clustering analysis led to the formation of seven groups by Charrad et al. [35] criterium (Fig 5). Group I contained 24 accessions, most of which were developed by the International Center for Tropical Agriculture (CIAT) and included accessions with different seed colors. Group II contained 28 accessions, including accessions from CIAT, IDR–Paraná, and the Brazilian Agricultural Research Corporation (EMBRAPA). Group III contained the greatest number of accessions ($n = 48$), including 33 of the 47 IDR–Paraná improved lines included in the study. Group was the smallest ($n = 17$) and mostly included carioca and black accessions. Group V contained 34 accessions, the majority of which had been developed by the EMBRAPA. Group VI contained 38 accessions, most of which were obtained from the Instituto Agronômico de Campinas (IAC) and IDR–Paraná and included accessions with different seed color. Group VII contained 32 accessions, most of which, like in the sixth group, were obtained from the IAC and IDR–Paraná programs.

In general, the CIAT accessions were separated from the EMBRAPA, IAC, and IDR–Paraná accessions. However, most of the lines from IDR–Paraná were placed in groups with CIAT accessions. No correlation was observed between the distance matrices of the morpho-agronomic and molecular data.

## Discussion

The common bean is an important food crop and a primary component of diets of populations in Latin America and in Eastern and Southern Africa [37]. In this context, the development of cultivars that combine broad adaptation, resistance and/or tolerance to biotic and abiotic stresses, nutritional quality, and high-yield potential is of utmost importance, and gene banks play an important role as depositories of desirable traits that can be used in breeding programs. The present study evaluated a panel Mesoamerican bean accessions that included elite

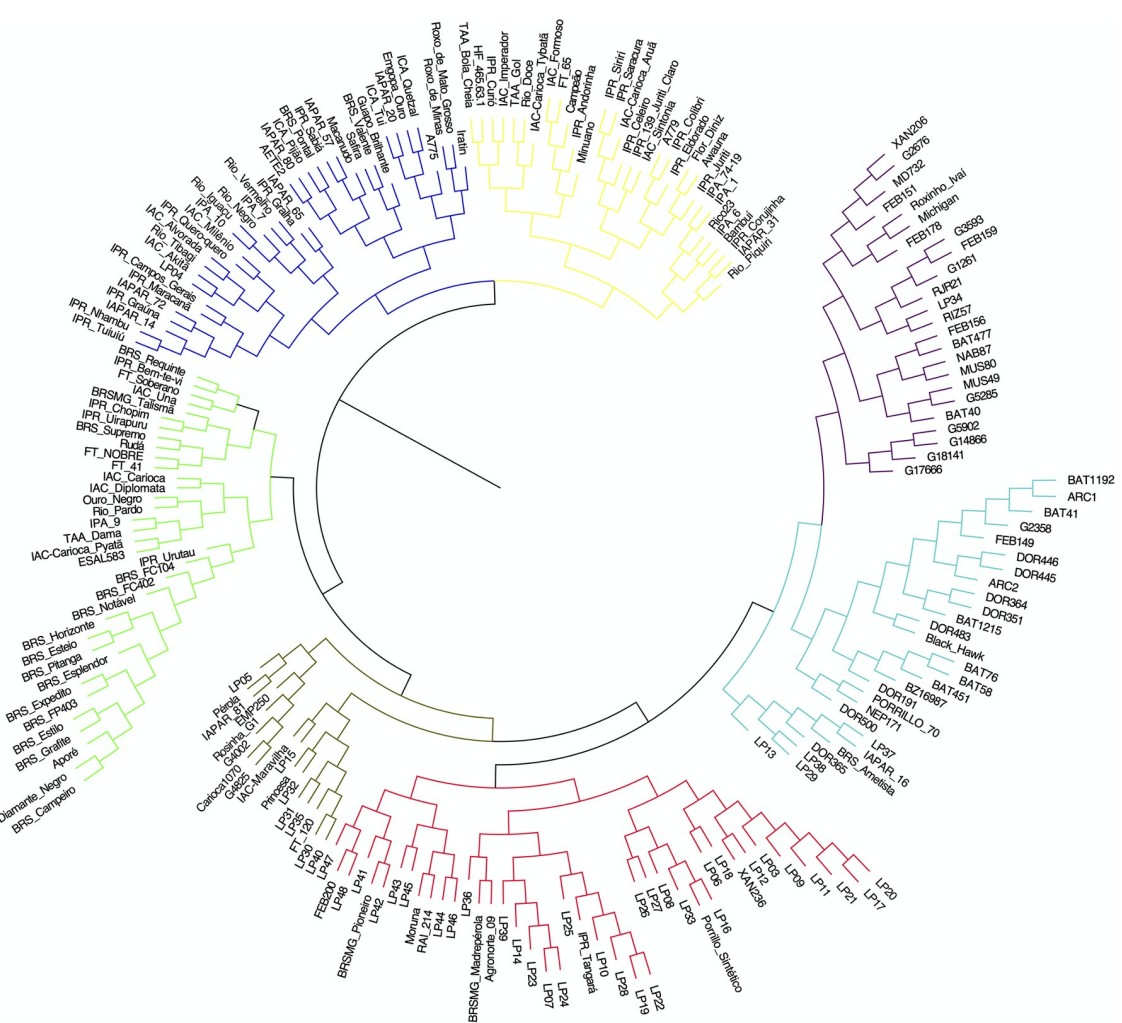

**Fig 5. Neighbour-joining cluster analysis of 215 Mesoamerican common bean accessions.** Analysis was performed using Jaccard's genetic distance.

cultivars, obsolete cultivars, lines from the IDR–Paraná breeding program, and landraces. The panel represents a wide diversity of germplasm of Mesoamerican origin, and the morpho-agronomic, biochemical, and molecular characterization of the panel has provided important information for maximizing its conservation and for utilizing its genetic material in breeding programs [38].

The wide variation observed in the morpho-agronomic and biochemical traits of the MPBD indicates that it is possible to select promising genitors for use in breeding programs that aim to improve traits related to plant architecture, seed yield, and nutritional quality. According to the criteria proposed by Rezende and Duarte [39], the Ac values indicated that experimental accuracy of the present study was very high ($\geq$0.90) for DPPH, TFC, TPC, W100, and NN, anc high ($\geq$0.70) for the other traits, except YLD, which was low. Ac values indicate the accuracy of the inferences of the low Ac value obtained for YLD is a consequence of the high influence of the environment for the trait, making it difficult to select the best accessions for selection purposes and about the effectiveness of inferences about accession genotype, and is a correlation between the predicted and the true genotypic values [16]. The

low Ac value obtained for YLD is a consequence of the high influence of the environment for the trait, making it difficult to select the best accessions. Delfini et al. [40] estimated genetic parameters for 39 Brazilian bean cultivars and obtained a lower $h^2$ for YLD (0.12) than for other traits.

The absence of correlation between YLD and the other morpho-agronomic or biochemical traits is considered a complicating factor in the selection of genitors. Delfini et al. [40] observed that YLD was correlated with traits related to seed morphology in black-type common bean, but with the number of seeds produced in carioca-type. These facts are related to the objectives of breeding programs in Brazil. For carioca-type beans, breeding programs emphasize seed size (large), shape (oblong), and color (light beige with a light brown stripe), and seed size is inversely related to yield [41,42]. However, large seeds are not important in the black commercial group, so cultivars with smaller seeds and high yield potential have been selected [40].

A moderate correlation among biochemical traits can be related to the duration and type of post-harvest storage. Kibar and Kibar [43] evaluated changes in the nutritional, bioactive, and morphophysiological properties of beans stored with different levels of moisture and reported a high correlation between DPPH and TPC before storage (r = 0.94, $p < 0.01$), but not after storage. Several factors (e.g., temperature, relative humidity, storage duration, and seed characteristics) affect the nutritional quality and antioxidant activity of beans [44].

In the present study, the hierarchical clustering of morpho-agronomic and biochemical traits revealed that W100 and TPC were important for separating the accessions into groups. In general, the accessions could not be separated using seed color (black, carioca, or colored) or type of genetic material (cultivar, lines, or landraces). Group IV, which contained 27 accessions, was considered important regarding biochemical traits, and may be further investigated for its potential use in breeding programs that target crop nutrition.

AFLP analysis indicated the MPBD contained high genetic variation, and the levels of polymorphism were higher or like those reported by Perseguini et al. [45] and Blair et al. [46], who also used AFLP markers in *Phaseolus vulgaris* L. This difference may be related to accession diversity, primers choice, or use of automated capillary electrophoresis. In addition, the groups achieved by clustering analysis of molecular data did not correlate with the groups based on seed color or genetic material. In the present study, most of the accessions that came from the same breeding program were clustered in the same group, as previously reported by Arunga et al. [47].

Similar to previous studies [10,48,49], there was either no association or only moderate association between the topologies based on morpho-agronomic and biochemical data and those based on molecular data, which indicates that both types of characterization (phenotype and genotype) are important for understanding the differentiation between MPBD accessions. Several previous studies of common bean have indicated the importance of both phenotypic and molecular characterization for elucidating variability [50–52]. Leite et al. [53] emphasized that, when using gene bank accessions as sources of variability in genetic breeding programs, the choice of precursor strains should be based on both genotyping and phenotyping data, so as to meet the breeders expectations of high genetic value, heterosis, and genetic variability.

## Conclusions

The MPBD accessions demonstrated high variation for all the morpho-agronomic, biochemical, and molecular traits included in the present study. There were no correlations between the topologies based on morpho-agronomic and biochemical characterization and that based on molecular traits, which indicates that both datasets are important for elucidating the

differences between accessions. The groups observed in the present study facilitated the identification of the most promising accessions to be exploited by genetic breeding programs.

## Supporting information

**S1 Table. List of accessions constituting the Mesoamerican Panel of Bean Diversity (MPBD).**
(DOCX)

## Author Contributions

**Conceptualization:** Vania Moda-Cirino, Leandro Simões Azeredo Gonçalves.

**Data curation:** Vania Moda-Cirino.

**Formal analysis:** Alison Fernando Nogueira, Jessica Delfini, Luriam Aparecida Brandão, Leonel Vinicius Constantino, Douglas Mariani Zeffa, José dos Santos Neto.

**Investigation:** Alison Fernando Nogueira, Luriam Aparecida Brandão, Silas Mian, Leonel Vinicius Constantino, Douglas Mariani Zeffa, José dos Santos Neto.

**Methodology:** Jessica Delfini, Leonel Vinicius Constantino.

**Writing – original draft:** Alison Fernando Nogueira.

**Writing – review & editing:** Vania Moda-Cirino, Jessica Delfini, José dos Santos Neto, Leandro Simões Azeredo Gonçalves.

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
