## [Decision Letter · Decision Letter 0]

19 Jan 2021

PONE-D-20-35240

Morpho-agronomic, biochemical and molecular analysis of genetic diversity in the Brazilian Mesoamerican common bean panel

PLOS ONE

Dear Dr. Azeredo Gonçalves,

Thank you for submitting your manuscript to PLOS ONE. After careful consideration, we feel that it has merit but does not fully meet PLOS ONE’s publication criteria as it currently stands. Therefore, we invite you to submit a revised version of the manuscript that addresses the points raised during the review process.

We look forward to receiving your revised manuscript.

Kind regards,

Roberto Papa, PhD

Academic Editor

PLOS ONE

Additional Editor Comments:

The authors need to carefully revise their manuscript to develop a new manuscript on the basis of reviewers comments and suggestions

Reviewers' comments:

Reviewer's Responses to Questions

**Comments to the Author**

1. Is the manuscript technically sound, and do the data support the conclusions?

Reviewer #1: Yes

Reviewer #2: No

2. Has the statistical analysis been performed appropriately and rigorously? 

Reviewer #1: Yes

Reviewer #2: N/A

3. Have the authors made all data underlying the findings in their manuscript fully available?

Reviewer #1: No

Reviewer #2: Yes

4. Is the manuscript presented in an intelligible fashion and written in standard English?

Reviewer #1: Yes

Reviewer #2: No

5. Review Comments to the Author

Reviewer #1: First, I would like to highlight the importance and the need for studies involving germplasm characterization. In my opinion, this is the main stage of pre-breeding, especially when evaluations are not limited to morphological traits, as in the current study. In this sense, I would like to congratulate all authors, and the agronomy department of UEL and IAPAR, they have been developing great work for years on genetic breeding of bean crop.

General considerations:

The present study is about the evaluation and characterization of the genetic diversity of a set of 215 accessions, through morphological, biochemical, and molecular analyzes. A lot of work was done to obtain the data, and several analyzes were conducted. The study is of great importance for crop improvement; however, the discussion needs to be improved. The results could be much better explored, very few studies were cited, and the conclusion was generic.

Specific considerations are included in the attached file.

Reviewer #2: Manuscript number: PONE-D-20-35240

Title: Morpho-agronomic and molecular analysis of genetic diversity in the Brazilian Mesoamerican common bean panel

Comments:

• The bibliography needs to be completely revised (heavily). Each statement must be verifiable by the reader, for this, a precise reference to the sources is necessary. The form and style of the bibliographic citations are not uniform, and an inappropriate format is used. For instance, see references 18, 19, 24, 26, and 27.

• English and grammar are too poor. The science is not communicated with sufficient clarity, making ambiguous understanding for some parts of the manuscript. My recommendation is to have the article edited by a mother-tongue with excellent English-writing skills.

• The abstract does not meet the standard scientific requirements: here the results are presented in a disjointed ways and there is a lack of a general and effective view of the work.

• The accurate interpretation of the data obtained is completely absent. For this reason, the part of the discussion requires substantial changes aimed at breaking down and contextualizing the results achieved. Moreover, the introduction is not well argued: the concepts need to be connected with each other more fluently.

In light of these considerations, and taking into account the potential positive impact of this study in future breeding programs, I reject the article entitled “Morpho-agronomic and molecular analysis of genetic diversity in the Brazilian Mesoamerican common bean panel” with the possibility of re-submission, once the manuscript will be revised.

6. PLOS authors have the option to publish the peer review history of their article (what does this mean?). If published, this will include your full peer review and any attached files.

Reviewer #1: **Yes: **Caléo Panhoca Almeida

Reviewer #2: No

---

## [Author Response · Author response to Decision Letter 0]

15 Feb 2021

Dr. 

Editor-in-Chief

Plos One

Please find enclosed our response to the review about the manuscript entitled “Morpho-agronomic, biochemical and molecular analysis of genetic diversity in the Mesoamerican common bean panel” that we are re-submitting for consideration for publication at Plos One. 

We thank the Editor and the reviewers to taking the time to review our manuscript. All the suggestions were very important to the construction of our manuscript. We have accept all the suggestions directly in the text, and response the doubts in this letter. We are available for any further clarification.

We kindly thank you for your time and consideration,

Leandro SA Gonçalves

Professor Universidade Estadual de Londrina

 

Responses to Reviewer(s)' Comments:

Reviewer #1

ABSTRACT: The summary is well written; however, a main conclusion is missing. 

L35 – The word "heritability" is duplicated. 

L43 - Delete "AFLP".

Response: The suggestions were accepted.

INTRODUCTION The introduction is well written and covers the main topic of the study. However, there was a lack as to the importance of using the three types of analysis (Morpho-agronomic, biochemical, and molecular). 

Response: This information was inserted in the manuscript.

L57-L60 – The citation ([6]) does not support the first three arguments of the sentence (1° one of the world’s largest producers and consumers of beans; 2° Mesoamerican cultivars are more popular; 3° carioca and black commercial types are preferred, accounting for 85 and 11% of Brazil’s bean production). Provide references. 

L67 – Change “variation and potential” by “diversity”. 

L67 – Change “characteristics” by “traits”. 

L70 – Change “such accessions must be fully characterized and evaluated” by “the candidate accessions must be fully evaluated and characterized, both morphologically and molecularly.” 

L71 – Change “exploitation” by “exploration”. 

L71-74 – Maybe would look better with a sentence like that (just a suggestion): “The precise exploration of genetic diversity by the breeding programs is necessary for the development of new commercial cultivars adapted to the most diverse regions of the country. Usually, higher levels of diversity in the set used for breeding is related to a greater chance of identifying higher agronomic traits [12, 13].” 

L74 – Change “[12] [13]” by “[12,13]”. 

L75- Change “The common bean exhibits wide variation in a variety of traits” by “The common bean exhibits wide agronomic traits variation”. 

L82- Change “various” by “several”. 

L83 – Change “[12] [16]” by “[12,16]”

Response: The suggestions were accepted.

MATERIAL AND METHODS

For a better understanding of the study, the authors should provide more details about the panel. Why were the panel accesses selected? What percentage of Cariocas, black, and colored beans? How many are commercial cultivars and how many are breeding lines?

Response: This information is included in Supplementary Table S1. 

Another important point, based on what information is it possible to assure that the selected accessions are Mesoamerican?

Response: Based on the characteristics of the seed and through the molecular analysis of accessions. This molecular characterization was carried out using the AFLP and SNPs markers. Delfini J, Cirino VM, Neto S, Ruas PM, César G, Ana S, et al. Population structure , genetic diversity and genomic selection signatures among a Brazilian common bean germplasm. Scientific Reports. 2021; 1–12. doi:10.1038/s41598-021-82437-4

The description of the “Morpho-agronomic characterization” section could be improved. Were all evaluations performed after the harvest? How many plants from the useful plot were evaluated?

Response: This information was inserted in the manuscript.

L94 – add “from different breeding programs and countries.” 

L97 – add the sowing month of the experiment. 

L105 – Change “([18])” by “[18]”. 

L153 – Change “Amplified fragment length polymorphism (AFLP)” by “AFLP”, the abbreviation has already been defined in the summary.

Response: This information was inserted in the manuscript.

RESULTS

The results are clear and well presented. 

The values of Ac and h2 (YLD) need to be revised, figure 1 shows one value and the text describes another. Another point, the abbreviation “FT (total phenolic contents)” is like “TP” in figures 1, 2 and 3. 

Response: Abbreviations have been modified in the manuscript.

L222 – Change “The Ac values were high (≥ 0.70) for most traits but relatively low (0.52) for YLD (Figure 1).” by “The Ac values were high (≥ 0.70) for all traits, except for YLD (0.52)”. 

L223 – Change “Meanwhile, h2 was high for DPPH, FT,224 FLA, W100, and NN, low for YLD (0.27), and intermediate (0.59–0.68) for all other traits” by “Meanwhile, h2 was high (0.98 – 0.90) for DPPH, FT, FLA, W100, and NN, intermediate (0.59–0.79) for all other traits, except for YLD (0.15).” 

L250 – “of the selected accessions” Which accessions were selected? Or does it refer to the panel? 

L268 and L272 – The IFP “cm” unit is missing. 

L309 and 335 – Add more details to the legend (e.g., the traits used for analysis, grouping method, and the colors of the figure)

Response: The suggestions were accepted.

DISCUSSION

L345 – The panel was not “established”. Change by “evaluated”. 

L355 – Change “for DPPH, FLA, FT, W100, and NN, high ( 0.70) for SP, PL, IFP, and STL, and low for YLD.” by “and high ( 0.70) for the other traits, except YLD, which was low.” 

361 – Change “The low Ac value obtained for YLD may hinder the selection of accessions. The variable was highly influenced by the environment and was reflected in the heritability” by “The low Ac value obtained for YLD is a consequence of the high influence of the environment for the trait, making it difficult to select the best accessions.” 

362 – add “also” after “et al. [40]”. 

L386 – Change “PerseguinI’ by “Perseguini” 

L394, 407 and 409 – “the word "topologies" sounds strange. 

L410 – Change “for future genetic breeding programs” by “to be exploited by genetic breeding programs.”

Response: The suggestions were accepted.

REFERENCES:

Response: References were checked and corrected.

SUPPORTING INFORMATION

Response: The suggestions were accepted.

Reviewer #2

 The bibliography needs to be completely revised (heavily). Each statement must be verifiable by the reader, for this, a precise reference to the sources is necessary. The form and style of the bibliographic citations are not uniform, and an inappropriate format is used. For instance, see references 18, 19, 24, 26, and 27.

Response: References were checked and corrected.

English and grammar are too poor. The science is not communicated with sufficient clarity, making ambiguous understanding for some parts of the manuscript. My recommendation is to have the article edited by a mother-tongue with excellent English-writing skills.

Response: The manuscript was reviewed for the English language to eliminate possible grammar or spelling errors.

The abstract does not meet the standard scientific requirements: here the results are presented in a disjointed ways and there is a lack of a general and effective view of the work.

Response: The abstract has been corrected.

The accurate interpretation of the data obtained is completely absent. For this reason, the part of the discussion requires substantial changes aimed at breaking down and contextualizing the results achieved. Moreover, the introduction is not well argued: the concepts need to be connected with each other more fluently.

Response: The article was revised for a better understanding of the readers.

---

## [Decision Letter · Decision Letter 1]

26 Mar 2021

Morpho-agronomic, biochemical and molecular analysis of genetic diversity in the Mesoamerican common bean panel

PONE-D-20-35240R1

Dear Dr. Azeredo Gonçalves,

We’re pleased to inform you that your manuscript has been judged scientifically suitable for publication and will be formally accepted for publication once it meets all outstanding technical requirements.

Kind regards,

Roberto Papa, PhD

Academic Editor

PLOS ONE

Additional Editor Comments (optional):

Reviewers' comments:

Reviewer's Responses to Questions

**Comments to the Author**

1. If the authors have adequately addressed your comments raised in a previous round of review and you feel that this manuscript is now acceptable for publication, you may indicate that here to bypass the “Comments to the Author” section, enter your conflict of interest statement in the “Confidential to Editor” section, and submit your "Accept" recommendation.

Reviewer #1: All comments have been addressed

Reviewer #2: All comments have been addressed

2. Is the manuscript technically sound, and do the data support the conclusions?

Reviewer #1: Yes

Reviewer #2: Yes

3. Has the statistical analysis been performed appropriately and rigorously? 

Reviewer #1: Yes

Reviewer #2: N/A

4. Have the authors made all data underlying the findings in their manuscript fully available?

Reviewer #1: Yes

Reviewer #2: Yes

5. Is the manuscript presented in an intelligible fashion and written in standard English?

Reviewer #1: Yes

Reviewer #2: Yes

6. Review Comments to the Author

Reviewer #1: I would like to thank the authors for their detailed responses toward my concerns and congratulate Gonçalves LSA and your team for the work they have been doing in the genetic breeding of common beans, especially for the topics related to nutritional crop improvement.

I still think that the study's discussion could be better explored, however, the authors made some changes.

Once again, congratulations on your study!

Reviewer #2: The manuscript entitled “Morpho-agronomic and molecular analysis of genetic diversity in the Brazilian Mesoamerican common bean panel” has been revised and implemented in accordance with the previous requests.

7. PLOS authors have the option to publish the peer review history of their article (what does this mean?). If published, this will include your full peer review and any attached files.

Reviewer #1: **Yes: **Caléo Panhoca Almeida

Reviewer #2: No

---

## [Editor Report · Acceptance letter]

12 Apr 2021

PONE-D-20-35240R1 

Morpho-agronomic, biochemical and molecular analysis of genetic diversity in the Mesoamerican common bean panel 

Dear Dr. Azeredo Gonçalves:

I'm pleased to inform you that your manuscript has been deemed suitable for publication in PLOS ONE. Congratulations! Your manuscript is now with our production department. 

Kind regards, 

on behalf of

Prof. Roberto Papa 

Academic Editor

PLOS ONE